Using ForeStereo and LIDAR data to assess fire and canopy structure-related risks in relict Abies pinsapo Boiss. forests

http://orcid.org/0000-0001-5350-9028 Cortés-Molino Álvaro 1 alvarocm@uma.es
Aulló-Maestro Isabel 2 3
Fernandez-Luque Ismael 1
http://orcid.org/0000-0002-0940-4541 Flores-Moya Antonio 1
Carreira José A. 4
http://orcid.org/0000-0002-1464-9770 Salvo A. Enrique 1
1 Departamento de Botánica y Fisiología Vegetal, Universidad de Málaga , Málaga , Spain
2 Escuela Técnica Superior de Ingeniería de Montes, Forestal y del Medio Natural, Universidad Politécnica de Madrid , Madrid , Spain
3 Departamento de Silvicultura y Gestión Forestal, Instituto Nacional de Investigación y Tecnología Agraria y Alimentaria, Centro de Investigación Forestal (INIA-CIFOR) , Madrid , Spain
4 Departamento de Biología Animal, Biología Vegetal y Ecología, Universidad de Jaén , Jaén , Spain
Huang Cho-ying
Electronic publication date: 2020 Oct 22
Publication date: 2020
Volume: 8
Electronic Location ID: e10158
Received 2020 Mar 3; Accepted 2020 Sep 21
Copyright: © 2020 Cortés-Molino et al.
Copyright year: 2020
Copyright holder: Cortés-Molino et al.
License: This is an open access article distributed under the terms of the Creative Commons Attribution License, which permits unrestricted use, distribution, reproduction and adaptation in any medium and for any purpose provided that it is properly attributed. For attribution, the original author(s), title, publication source (PeerJ) and either DOI or URL of the article must be cited.
License URL: https://creativecommons.org/licenses/by/4.0/

Keywords: Fir forests, Hemispherical images, Remote sensing, Forest conservation, Airborne LIDAR, Fire simulations

Funding: The authors received no funding for this work.

==============================
In this study we combine information from aerial LIDAR and hemispherical images taken in the field with ForeStereo—a forest inventory device—to assess the vulnerability and to design conservation strategies for endangered Mediterranean fir forests based on the mapping of fire risk and canopy structure spatial variability. We focused on the largest continuous remnant population of the endangered tree species Abies pinsapo Boiss. spanning 252 ha in Sierra de las Nieves National Park (South Andalusia, Spain). We established 49 sampling plots over the study area. Stand structure variables were derived from ForeStereo device, a proximal sensing technology for tree diameter, height and crown dimensions and stand crown cover and basal area retrieval from stereoscopic hemispherical images photogrammetry. With this information, we developed regression models with airborne LIDAR data (spatial resolution of 0.5 points∙m−2). Thereafter, six fuel models were fitted to the plots according to the UCO40 classification criteria, and then the entire area was classified using the Nearest Neighbor algorithm on Sentinel imagery (overall accuracy of 0.56 and a KIA-Kappa Coefficient of 0.46). FlamMap software was used for fire simulation scenarios based on fuel models, stand structure, and terrain data. Besides the fire simulation, we analyzed canopy structure to assess the status and vulnerability of this fir population. The assessment shows a secondary growth forest that has an increasing presence of fuel models with the potential for high fire spread rate fire and burn probability. Our methodological approach has the potential to be integrated as a support tool for the adaptive management and conservation of A. pinsapo across its whole distribution area (<4,000 ha), as well as for other endangered circum-Mediterranean fir forests, as A. numidica de Lannoy and A. pinsapo marocana Trab. in North Africa.

Introduction

Endemic conifer species are more numerous in Mediterranean-type climate regions in the Northern Hemisphere than in the Southern Hemisphere, which has been linked to the selective pressure of cold and/or drought conditions that led to the development of ecophysiological advantages for conifers over angiosperms on oligotrophic soils. Meanwhile, the Mediterranean-type climate regions of South Africa and Southwestern Australia have been more climatically and tectonically stable, which resulted in lower diversity and persistence of ancient lineages of conifers. The Mediterranean Basin has 32 endemic conifer species, accounting for more than 25% of the total conifer flora of 122 species (Rundel, 2019). In particular, the genus Abies Mill. experienced extensive speciation from the late Neogene that gave rise to nine species and one natural hybrid in the Mediterranean Basin (Linares, 2011). Past climate changes have led to population migrations, and to shrinkage and fragmentation of ancestral Mediterranean fir populations, further exacerbated by human impacts. This resulted in circum-Mediterranean endemic firs of high paleogeographic interest, since they are established in relict restricted-range populations with relevant vulnerability to global warming effects (Liepelt et al., 2010). Adaptive management of these forests to protect them from the increasing fire risk is essential for their survival.

Extreme climate events, such as severe droughts, mega-fires, and disease infestations threaten these relict Mediterranean fir populations (Sánchez-Salguero et al., 2017). It is well known that fire has influenced the landscape and terrestrial life as far back as the beginning of land plants (Bowman et al., 2009; Pausas & Keeley, 2009; He et al., 2012). Although many conifers have developed adaptive traits to live in fire-prone environments, this is not the case for the genus Abies. The firs developed traits appropriate for the humid areas where they thrive, which has rendered them neither resistant (thin bark) nor resilient (recruitment failure in open spaces) to fires (Furyaev, Wein & MacLean, 1983; Vega Hidalgo, 1999).

Remote sensing is useful for assessment and development of measures for mitigation of the effects of global warming in relict Mediterranean fir forests. Spectral imagery has been employed for the early detection of forest pathogen infestations (Immitzer & Atzberger, 2014), to estimate evapotranspiration (Dzikiti et al., 2019), and to study photosynthetic activity (De Sousa et al., 2017). Meanwhile, 3D point cloud data from laser scanning (LIDAR) have been employed in fire management (Chuvieco & Kasischke, 2007) and to assess forest volume, biomass (Van Ardt, Wynne & Scrivani, 2008), and canopy structure (Adamic et al., 2017; Mura et al., 2015). Also, the point cloud can be used for ecological purposes, such as assessing light availability for species distribution modeling (Wüest et al., 2020) and forest changes in ecotones (Wang, Ginzler & Waser, 2020). Airborne LIDAR has shown better suitability for mapping crown and canopy heights (Wang & Glenn, 2008), although in high density forests the point cloud may not reach the ground, and thus mapping understory vegetation may be inaccurate. However, terrestrial LIDAR has a great potential for estimating shrub and understory biomass, although there are insufficient points for a precise estimation of crown heights when the canopy cover is high (Hilker et al., 2012).

Mapping fire risk with the support of remote sensing tools is becoming essential for landscape planning in the Mediterranean Basin. High-precision fuel moisture and flammability spatial modeling is achieved by combining satellite and meteorological data into radiative transfer models (Chuvieco et al., 2006; Yebra et al., 2018). Burn probability is then assessed through algorithms such as the Minimum Travel Time (MTT) based on the Huygens’ principle (Finney, 2002). Several studies have previously applied MTT through FlamMap software on fuel spatial models to assess fire risk in Mediterranean-type ecosystems of Greece (Mitsopoulos, Mallinis & Arianoutsu, 2015; Mallinis et al., 2016), Italy (Salis et al., 2015) and Spain (Molina, Rodríguez y Silva & Herrera, 2017; Alcasena et al., 2019). In this last study, fire risk and highly vulnerable areas were mapped for the whole Catalonia region by applying the Scott & Burgan (2005) fuel model classification on vegetation structure data and running MTT through FlamMap to obtain 150 m resolution fire scenarios. Alternatively, fire spread from specific ignition events can be forecasted, for example, Salis et al. (2016) used the FARSITE software to derive fire spread simulations for several Euro-Mediterranean countries along an east-west gradient. All these studies agree that accurate and customized fuel models are key for assessing burn probability and fire risk.

In this respect, airborne LIDAR technology provides an unprecedented tool for fuel and canopy structure characterization in forest ecosystems. However, several studies highlighted limitations of this technology for accurate understory fuel mapping due to the lack of points reaching the ground (González-Olabarria et al., 2012; Botequim et al., 2019). Therefore, LIDAR data need to be implemented in regression models supported by field sampling to eventually characterize the forest structure. For this purpose, hemispherical images are an alternative to traditional field sampling. This technique has been used in forest ecology for more than 50 years, but its widespread adoption was limited due to constraints related to image processing capacity (Chianucci, 2019). However, technical improvements allowed reducing the time for image processing as well as better image quality acquisition. The widespread proliferation of digital cameras has increased the ease of obtaining and storing hemispherical images, which have become an important tool for fieldwork (Hall, Fournier & Rich, 2017). ForeStereo, a forest inventorying device patented by the Forest Research Centre of the Spanish National Institute for Agriculture and Food Research and Technology (CIFOR-INIA), allows one to obtain stand and tree variables in a cost-effective way by processing pairs of stereoscopic hemispherical images taken at a sampling location (Rodríguez-García et al., 2014).

Most studies applying LIDAR to circum-Mediterranean fir forests have focused on the most widely distributed Abies alba Mill., whereas those focusing on other species such as the relicts A. pinsapo Boiss. and A. numidica de Lannoy, which are becoming increasingly vulnerable to global change impacts (Liepelt et al., 2010), are very scant. Aragón, Navarro Cerrillo & Mesas Carrascosa (2019) and Cortés-Molino et al. (2017) studied A. pinsapo Boiss. forests using LIDAR, but only for basic tree identification and vegetation landscape analysis, respectively. Now, the combination of remote sensing technology such as laser scanning and proximal sensing (e.g., ForeStereo) can contribute to the monitoring of these relict forests through the acquisition of high-precision stand structure data.

Abies pinsapo is restricted to a few areas in the southern Iberian Peninsula (A. pinsapo pinsapo) and northern Morocco (Abies pinsapo marocana), totaling less than 8,000 ha (Linares, 2008). Forest fires have markedly reduced the size of populations of this fir; in some localities the longest timespan without fires in the period 1817–1997 was just 34 years (Vega Hidalgo, 1999). Thus, fire is considered the most important threat for the conservation and survival of this endangered species (López-Quintanilla et al., 2013). A. pinsapo shows a very low resistance to fire due to its thin bark, despite its relatively low fuel flammability and low fire spread rates in dense stands, due to sparse understory and relatively humid conditions (Rodríguez y Silva, 1996). Additionally, acute symptoms of tree growth decline and forest dieback due to stand stagnation and climate change have already been reported in some populations (Linares & Carreira, 2009), where Pinus halepensis Mill. is increasing in abundance, turning previously pure A. pinsapo stands into mixed ones (Linares, Delgado-Huertas & Carreira, 2011).

Our work aimed to combine the use of LIDAR and hemispherical images to study one of the most significant A. pinsapo populations, located in a protected area in Málaga (Spain), to assess vulnerability through (i) mapping fire risk and (ii) analyzing canopy structure variability and its possible links to reported declining growth symptoms.

Materials and Methods

Study area

The study location is a steep valley of about 250 ha in area in the municipality of Yunquera, in Sierra de las Nieves National Park (Fig. 1), in a transition between the upper and lower Mesomediterranean bioclimatic band. The annual rainfall is around 1,500 mm and the average daily maximum temperature of the warmest month (August) is 33.6 °C (Rivas-Martínez & Rivas-Sáenz, 1996). At the southern border of the valley there is a crest that was the limit of a severe wildfire in 1991 that burned 9,000 ha (Narváez, 1991). The eastern part is bordered by crop fields. This, together with summer weather conditions and significant touristic pressure in Sierra de las Nieves National Park, makes the risk of wildfire especially high. This forest belongs to the Paeonio broteroi-Abietetum pinsapo (Asensi & Rivas-Martínez, 1976) vegetation association composed mainly of pinsapo fir, forming single-species stands in the upper and shaded parts of the valley. The incidence of the root-rot fungus Heterobasidion abietinum Niemelä & Korhonen is very high (Linares et al., 2010). In sunny and low-altitude spots, the forest includes Pinus halepensis and shrubs of Juniperus spp and Cistus spp.

Figure 1 Study area location, in the National Park of Sierra de las Nieves (Andalusia, Spain).

Forty-nine plots were used in the field-based sampling for calibration purposes using ForeStereo device. Yellow points are training plots, red ones are validation plots. White arrows indicate the firewalls opened on the area. Base map and data from OpenStreetMap and OpenStreetMap Foundation.

Fieldwork: ForeStereo inventory

The purpose of the fieldwork was to classify local fuel models and collect forest structure data, mainly canopy cover, and crown and stand heights. The valley was sampled in Spring 2018 with 49 plots of 8 m radius (201.06 m2), using stereoscopic hemispherical images for obtaining tree metrics such as stand height (Ho), Canopy Base Height (CBH), Canopy Cover (CC), Canopy Bulk Density (CBD) and basal area (G). Shrub cover and height were also assessed by the line-intersect method to support the classification of the fuel models. Each sampling plot was accurately geolocated using a high precision GNSS receiver, supported by an RTK terrestrial station deployed in the upper part of the valley. We assumed a maximum error of 1 m in each plot, due to the difficulty of getting GNSS coverage in high dense canopy.

Access to field sites was approved by the Andalusian Regional Government (Consejería de Medio Ambiente y Ordenación del Territorio) with the approval code: PNSN/AU/10-2018.

Forest inventory was derived using ForeStereo, a device developed by the Forest Research Centre of the Spanish National Institute for Agriculture and Food Research and Technology (INIA-CIFOR). ForeStereo is equipped with two upward-looking fish-eye cameras. At each sampling location three pairs of hemispherical stereoscopic images (Fig. 2) with different exposures are taken. The matching process, compiled in a MatLab® software package, consists of four main steps as detailed in Sánchez-González et al. (2016): (i) a supervised segmentation of tree stems and crowns; (ii) correspondence of features between the two images and photogrammetric retrieval of tree dimensions; (iii) tree variable modeling and (iv) stand variable estimation, which requires correction of instrumental bias and occlusions (Montes et al., 2019). ForeStereo was used to estimate tree height, crown base height, crown volume and diameter at breast height (DBH) for each tree, number of stems per hectare and crown cover (CC) and stand basal area (G). Tree metrics from the hemispherical images were compared with airborne LIDAR output data to develop regression models. Thirty-two of the plots were randomly chosen to adjust the models, and the remaining seventeen plots were used to assess the models’ predictive capability.

Figure 2 Example of hemispherical pair images obtained with ForeStereo to characterize forest stand canopy structure at the field sampling plots.

(A) and (B) are both stereoscopic images obtained at the same time in each plot.

Because the geometry of the ForeStereo system and image projections is known, no additional data calibration is needed to carry out photogrammetric retrieval of tree variables. Accuracy of ForeStereo estimated through the Root Mean Squared Error (RMSE) ranges from 0.015 to 0.057 m for DBH (Rodríguez-García et al., 2014; Sánchez-González et al., 2016), 2.59 to 6.4 m for tree height (Rodríguez-García et al., 2014; Marino et al., 2018) and 3.1 and 0.6 m for crown base height and crown diameter respectively (Marino et al., 2018), and 11.6 m2/ha for G (Sánchez-González et al., 2016).

With the ForeStereo data we were able to estimate stand height at each plot following the Assmann’s criteria (Assman, 1970), whereas the canopy cover was obtained directly from the hemispherical images, and the Canopy Base Height (CBH) was averaged for each plot. The calculation of the Canopy Bulk Density (CBD) was based on equations reported by Ruiz-Peinado, Del Río & Montero (2011) for A. pinsapo, in which tree height and stem diameter are used to calculate thin branch and needle biomass. Therefore, CBD is estimated by dividing biomass by crown volume (which is obtained from ForeStereo estimates).

LIDAR data

The LIDAR data were obtained in 2015 by the Spanish National Geographic Institute, through the “Plan Nacional de Ortofotografía Aérea (PNOA)” project. The point cloud density is 0.5 points∙m−2. FUSION software was employed for point cloud processing and data extraction (McGaughey & Carson, 2003). A correlation matrix between ForeStereo tree data and all LIDAR metrics obtained with FUSION was useful to detect which LIDAR metric was most suitable to build the regression models in 32 random plots. We selected the best correlation results (R < 0.5, p-value < 0.05) to test linear, power and exponential regression models. The models with less root-mean-square error (RMSE) and higher adjusted-R in the remaining 17 plots were chosen to predict ForeStereo tree metrics from the LIDAR point cloud. Height break for LIDAR metrics was 4 m, whereas it was 0.25 m for the total vegetation height to avoid high shrubs influence and ground points, respectively. These two height breaks were tested to inquire which one was the better to fit the models.

General canopy structure traits such as canopy cover and height were analyzed from the regression models obtained to detect symptoms of declining growth in this A. pinsapo forest. These variables were chosen because airborne LIDAR can estimate them accurately (Ahmed et al., 2015; Arumãe & Lang, 2018). Also, we tested whether canopy bulk density was consistent with the results from this analysis.

Fuel models and fire scenarios

To simulate fire risk, we first classified field plots according to the UCO40 fuel models, which use specific criteria and traits appropriate for Mediterranean environments and thus perform better than the widely used Prometheus or Rothermel models (Rodríguez y Silva & Molina Martínez, 2010, 2012). We identified a total of six fuel models across the plots (Table 1; Fig. 3). The UCO4 procedure is based on the fuel models classification of Scott & Burgan (2005), but adapted for southern Iberian Peninsula climate conditions through providing hybrid model types that represent fuel traits and their evolution. Shrub and climate characteristics have shown different behaviors between American and Mediterranean fuel models, so parameters such as fuel load and fuelbed depth must be adjusted (Rodríguez y Silva & Molina-Martínez, 2012).

Table 1 Fuel model classification obtained following the UCO40 criteria (Rodríguez y Silva & Molina-Martínez, 2012), based on Scott & Burgan (2005).

Dead fuel models are classified by the time lag: the time required for the moisture content of a fuel to respond to within 2/3 of the new equilibrium moisture content. Larger diameter fuels have longer time lags, so they respond slower to environmental changes (Anderson, 1982). LiveH: Live herbaceous fuel, LiveW: Live wood fuel. Moisture of extinction (%): moisture content that prevents flame from propagating.

Fuel type	Fuel loading (tn/ha)	Moisture of extinction (%)	Fuel bed depth (cm)	
Time lag	LiveH	LiveW	
1 h	10 h	100 h	
Predominance of shrubs	
M3	11.47	2.88	3.37	0	6.10	15	82.29	
M8	11.23	6.10	3.47	0	7.27	30	121.92	
M9	34.71	9.86	4.93	0	18.89	15	182.88	
Pine-needle litter with shrubs and/or grassland under forest canopy	
HPM4	17.63	13.23	1.17	0	11.13	20	76.2	
Predominance of pine-needle with branches and other canopy debris	
HR7	0.73	3.76	3.47	0	0	25	18.28	
Predominance of canopy debris accumulation	
R4	1.57	5.16	6.29	0	0	25	82	

Figure 3 Example of the six fuel models identified in the plots based on Rodríguez y Silva & Molina Martínez’s (2010) handbook.

(A) R4 model: Predominance of canopy debris accumulation (B) HR7 model: Predominance of pine needle with branches (C) HPM4 model: Pine-needle with shrubs and grass (D) M3 model: Predominance of shrubs with a fuel bed close to 1 m (E) M8 model: Predominance of shrubs with grass with a fuel bed of 120 m. (F) M9 model: Predominance of thick shrubs with a fuel bed over 175 m.

Ecognition® software was used to segment the place of study using the Nearest Neighbor algorithm in an Object-Based Image Analysis (OBIA) (Gao et al., 2007). To carry out this segmentation, we used the raster layers resulting from the regression models validated previously (CBD, CBH, G, Ho) along with NDVI data from Sentinel-2 images (2015) and terrain models obtained from the LIDAR point cloud (topography, aspect, and slope). Later, a confusion matrix was calculated to evaluate the accuracy of the fuel model classification.

The following raster layers generated from the regression models were incorporated: terrain elevation, aspect, slope, Ho, CBH, CBD, CC, and fuel models.

The initial fuel moisture file (.FMD) used was based on Scott & Burgan (2005) suggestions. We assumed a low moisture content (two-third cured) for the fuel more commonly found in north-facing slopes (fuel models M8, HR7 and R4; see Table 1) and very low moisture content (fully cured) for the fuel models more frequently found in south-facing slopes (M3, M9 and HPM4). A Weather Stream file (.WXS) with typical values of circadian change under summer weather conditions in the area was used for dead fuel moisture conditioning (Table 2). This file modifies initial dead fuel moisture based on changes during a given period in weather variables such as temperature, relative humidity, cloud cover and hourly precipitation. Conditioning also implies adjusting initial dead fuel moisture to site factors (elevation, slope, aspect, and canopy cover), based on the corresponding raster layers previously uploaded in FlamMap (Finney, 2006). The .WXS file was built from meteorological data that are continuously recorded in situ by the University of Jaén (values of temperature, precipitation and relative humidity) and from records of the Spanish Meteorological Agency-AEMET (values of wind and cloud cover). We chose the warmest day of 2014 (available recorded data) for the conditioning period between the 10:00 and 19:00 h without precipitations or any cloud cover. The conditioned fuel moistures at the end of this period were the final fuel moistures used for the simulations.

Table 2 Dead fuel moisture conditioning.

Weather stream file (.WXS) showing typical summer weather circadian change in the area, used as input to Flammap software for quantifying the moisture of dead fuel.

Date	T (°C)	RH (%)	PP (mm)	Wind SP (m/s)	Wind dir. (°)	Cloud (%)	
08/27/14 10:00	28	33	0	2	180	0	
08/27/14 11:00	29	32	0	2	180	0	
08/27/14 12:00	30	29	0	1	180	0	
08/27/14 13:00	31	27	0	2	180	0	
08/27/14 14:00	30	27	0	2	180	0	
08/27/14 15:00	29	28	0	2	180	0	
08/27/14 16:00	29	29	0	2	180	0	
08/27/14 17:00	28	30	0	1	180	0	
08/27/14 18:00	27	31	0	2	180	0	
08/27/14 19:00	26	32	0	2	180	0	

We obtained three different datasets as model outputs: (1) Burn probability based on 200 random ignition points using the MTT algorithm, (2) flame length and (3) flame spread rate, calculated for each cell. The purpose of simulating fire scenarios was to detect vulnerable areas and to assess for conservation planning how exposed the pinsapo forest is to this risk. For this reason, we did not simulate specific events or spotfires using Farsite software. Instead, FlamMap software is more appropriate, because it calculates spread rate and flame length for each landscape cell without a temporal component, and uses MTT to simulate 200 random fires to predict the probability of a point to be burned (Finney, 2006; González-Olabarria et al., 2012). The simulations were set under two prevailing wind conditions: west winds (locally called “Ponent”) and east winds (called “Levant”), both for a typical speed of 13 km∙h−1.

Results

We found that the LIDAR metrics that best fit the ForeStereo data were (Table 3): Percentage of first returns above 4 m (x), Percentage of all returns above 4 m (y), Percentage of all returns above mean (z) and Percentage of first returns above 0.25 m (d). All the significant regression models were obtained with 95% confidence in the seventeen validation plots. Basal Area (G) was the only variable with an acceptable fit using a height break of 0.25 m. The rest of the variables were better modeled above 4 m. Canopy Cover (CC) had the best fitting model with an RMSE less than 20%, while the greatest error was found in modeling the Canopy Base Height (CBH) with an RMSE of 83.3%. This high difference could be due to low point cloud density (minimum 0.5 points·m−2 guaranteed) as well as the fact that airborne LIDAR produces better accuracy for variables related to the top of the canopy (Hilker et al., 2012), such as canopy cover or canopy height, than variables under the canopy. Results for dominant height were acceptable (RMSE of 0.55), because finding the top of the crowns with ForeStereo in high-density forests can be difficult.

Table 3 Best regression models between LIDAR data (independent variable) and field-based Forestereo data (dependent variable), used to map spatial distribution of the main forest structure variables.

Dependent variable	Predictive model	R2	RMSE (%)	
CC	x∙0.92023	0.84	17.7	
Ho	0.1469∙y	0.61	55.7	
CBH	x0.013839	0.43	83.3	
CBD	z∙0.007893	0.64	59.5	
G	d∙9.28∙10−8	0.61	67.3	
Note:

To find which LIDAR data best suit to each field data, a correlation matrix was done. The best results (R > 0.5 and p-value < 0.05) were tested by using linear, power and exponential regression models. The models with highest R2 and lower RMSE were selected. CC: canopy cover; Ho: stand height; CBH: canopy base height; CBD: canopy bulk density; G: basal area. x: Percentage of LIDAR first returns above 4 m, y: Percentage of all returns above 4 m, z: Percentage of all returns above mean, d: Percentage of first returns above 0.25 m.

The error matrix for the fuel model classification using the Nearest Neighbor algorithm shows an overall accuracy of 0.56 and a Kappa Coefficient (KIA) of 0.46 (Table 4). The most frequent fuel model in the study area was M9 (30.5% of land cover), followed by M3 with 27.4% and M8 with 23%. In all the fuel models, shrubs play a predominant role in the fire behavior. The least frequent fuel models were HPM4 (7.19% of land cover; fire behavior mainly controlled by needle litter together with understory shrubs and/or grasses), HR7 (6.12% of land cover; conifer-needles and branches and other canopy debris play a predominant role) and R4 (5.73% of land cover; predominance of canopy debris accumulation in fire behavior).

Table 4 Confusion matrix of the Nearest Neighbor classification for the fuel models.

User accuracy: how often the class on the map will actually be present on the ground. Producer accuracy: how often are real features on the ground correctly shown on the classified map.

	Predicted classes	
		M3	M8	M9	HPM4	HR7	R4	Sum	
Actual classes	Fuel type observed in field	
M3	2	0	0	1	2	0	5	
M8	0	5	1	0	0	0	6	
M9	3	0	3	0	0	0	6	
HPM4	0	0	0	2	2	1	5	
HR7	0	1	0	0	0	1	2	
R4	0	0	0	0	0	3	3	
Sum	5	6	4	3	4	5		
Accuracy	
Producer	0.4	0.83	0.75	0.7	0	0.6		
User	0.4	0.83	0.5	0.4	0	1		
Overall accuracy	KIA	
0.56	0.46	

Once the corresponding layers were created based on the regression models, the fuel model classifications and the terrain elevation data (Fig. 4), fire simulations under two wind conditions were obtained from FlamMap (Fig. 5). The final landscape file keeps the same pixel resolution of the input data (10 m). We found higher burn probabilities and spread rate under Levant wind conditions, but similar flame length scenarios under both Levant and Ponient winds. Burn probability was higher under Levant winds, with a mean value of 0.078, than under Ponent winds, with a mean value of 0.061. Fire spread rate showed a mean value of 41.82 m∙min−1 under Levant wind conditions, and more than 50% of the landscape showed spread rates ≥50 m∙min−1. In contrast, Ponent winds resulted in lower fire spread rates (mean value of 33.34 m∙min−1) and a considerably lower fraction of the landscape (36%) was affected by spread rates ≥50 m∙min−1. Flame length values were similar for the fire simulations under both wind directions.

Figure 4 LIDAR raster layers produced to run Flammap.

(A) Canopy cover (%). (B) Stand height (m). (C) CBH (m). (D) CBD (Kg/m3). (E) Fuel models. (F) Aspect (°). (G) Elevation (m) (H) Slope (%).

Figure 5 Fire simulations obtained from Flammap for Ponent wind (West) and Levant wind (East) conditions.

(A) Burn probability in Ponent wind. (B) Spread rate in Ponent wind (m·min−1). (C) Flame length in Ponent wind (m). (D) Burn probability in Levant wind. (E) Spread rate in Levant wind (m/min). (F) Flame length in Ponent wind (m).

Regarding the canopy structure analysis to detect symptoms of forest decline and dieback, the estimated variables CBH and G were not considered due to RMSEs far above 60%. Instead, we use Ho and CC (Fig. 6), as well as CBD, for such assessment. The following results were estimated only in the landscape cells with a vegetation height above 4 m, in order to exclude non-forest patches of shrubs, as well as forest gaps recently opened due to tree mortality (Fig. 7). Canopy heights (Ho) ranged from 4 to 18 m, with a mean value of 9.1 m and a standard deviation (SD) of 3.28. It is remarkable that the mode of canopy heights falls below 5 m in this forest area not affected by fire since the mid-20th century and under long-term, no management policy. On the other hand, canopy height classes between 7 and 12 m showed similar frequencies (high equitability). Lastly, canopy heights higher than 15 m were present, but with rather low frequency.

Figure 6 Frequency histograms for canopy height (Ho, A) and canopy cover (CC; B) values, as derived from regressions between Forestereo and LIDAR data, in the studied A. pinsapo forest.

Figure 7 Photograph showing the widespread decline and dieback processes (stand stagnation, tree mortality, forest gap opening) that A. pinsapo forests are experiencing in the study area.

The die-back process of A. pinsapo forests stimulates the production of R4 fuel model (abundant dead wood and debris accumulation).

Canopy cover had a mean value of 64.5% and an SD of 18. We found low frequencies for values between 0% and 30% (<2% of the area) because most areas with low CC corresponded to land covered by shrubs with less than 4 m heights. Almost 25% of the land showed CC values above 80%, which means that areas with near to full cover are relatively common. Also, 66% of the area had a CC of 40–80%. Lastly, CBD mean value was 0.16 kg∙m−3 with a SD of 0.1. Our results showed a high canopy density because >60% of the area had values over 0.1 kg∙m−3 and >30% was over 0.3 kg∙m−3.

Discussion

Endangered circum-Mediterranean firs are highly vulnerable to climate change effects in the isolated areas where they remain (Sánchez-Salguero et al., 2017). Abies nebrodensis Mattei is currently the rarest conifer in the European flora, with only 34 mature trees able to reproduce sexually in the wild (Pasta et al., 2019). The recovery of this species and the protection of the other ones to avoid a similar decrease is an urgent matter that demands the best techniques available to support the traditional field survey.

We propose a methodology that combines the use of LIDAR with ForeStereo, UCO40 fuel models, and FlamMap simulations to significantly reduce the effort and time required for fieldwork, increasing the efficiency of the massive data capture required in forest management.

The application of this methodology in the study area obtained fire simulations that showed that east wind conditions (“Levant”) resulted in worse fire scenarios than west winds (“Ponent”) as illustrated in Fig. 5. Spread rate appears to be more influenced by topography and wind conditions (Salis et al., 2016) than by flame length, which appears to be more influenced by fuel characteristics. Low spread rate and flame length were found in areas with HR7 and R4 models because they correspond to high-density A. pinsapo stands, consistent with the findings of Rodríguez y Silva (1996).

Similar results can be found in the Euro-Mediterranean study of Salis et al. (2016), in which the maximum spread rates simulated in the Attica region (Greece), Budoni (Italy), and Fresnedoso de Ibor and Navalmoral (Spain) are between 50 and 110 m∙min−1. In their study, the worst flame length scenarios are located in Fresnedoso de Ibor (Spain) and Penteli (Greece) with a range between 25 and 50 m. They also found a higher spread rate and low flame length mainly in areas with herbaceous vegetation, but also in forest and shrublands in steep mountains exposed to wind (as in our study). In these areas flame length is also higher than in lands with herbaceous vegetation.

However, both wind conditions generate two remarkable foci of fire risk, well highlighted in the burn probability map (Fig. 5). One is in the north-west part of the valley, on south-facing slopes (180° N) where very dense and tall (>1.80 m) patches of the shrub Juniperus spp. on steep terrain represent ideal conditions for a high fire spread rate (>50 m∙min−1), whereas the flame length will depend more on the wind (Ponent >15 m, Levant >30 m). It is not surprising that this condition corresponds to the M9 fuel model (Table 1), in which massive shrub formations dominate the fire behavior. The other focus is in the eastern part of the valley, due to the occurrence of fuel models for which fire behavior is mainly determined by the combination of dense shrub cover and very steep slopes (>75°).

González-Olabarria et al. (2012) observed a lower fire risk landscape in a carefully managed even-aged forest of Pinus nigra. and P. pinaster, whereas our study corresponds to a non-managed forest of pinsapo firs. This contrast strengthens the argument for the urgent need for adaptive management of these endemic fir forests, abandoning the traditional paradigm of non-management in biological conservation. The kind of prevailing fuel model appears to be determinant for the fire scenarios obtained, and the current “don’t touch” management strategy, together with the invasion by shrubs into forest mortality gaps, seem to promote high fire risk fuel models in the area.

The distribution of canopy structure features depicted in Fig. 6 highlights: (i) that the most frequent stand height barely reaches 5 m, the mean value is just 9.1 m and figures higher than 12–15 m are rare despite the fact that A. pinsapo can reach up to 30–35 m in height (López-Quintanilla et al., 2013); (ii) that canopy cover has an average value of 64.5%, well below the full-cover criterion under a “set aside” and “no management” strategy since the late 1960s, and patches with 90–100% cover account for less than 10% of the whole area; and (iii) that there is an overall very high variability for both stand height and canopy cover values across the landscape, with a relatively high evenness in both variable distributions, especially regarding tree height. All these results indicate a lack of old-growth stands in the study area, and the predominance of secondary forests, which is consistent with previous studies based on field surveys (Linares, Carreira & Ochoa, 2011; Linares et al., 2013).

We found considerably high values of canopy density (CBD: Fig. 4), which can increase the risk of severe crown fires (Arellano Pérez et al., 2017). These canopy density values in a well below full-cover area, together with low Ho suggests two possible explanations: (i) the high CBD values correspond to full-cover patches with older stands where gaps are still not open and/or (ii) shrub strata are increasing their height above 4 m, interlocking with the canopy. Both are compatible with different phases of forest decline and the gap opening process.

The low stand height values we found, even in the patches with older stands and high cover values, could indicate symptoms of stand stagnation in such patches. A multi-temporal comparison (1957–2007) and fractal analysis of digital panchromatic aerial photographs of the same area (Linares et al., 2006; Linares, Camarero & Carreira, 2009), revealed a process of simultaneous stand densification and expansion of A. pinsapo at the landscape level in the last decades. This is a consequence of strict protection since the 1960s of an area mostly covered at the time by bare soils and open scrublands, with a few sparsely distributed and small stands and isolated trees of A. pinsapo. Increasing competition due to the densification of these regenerating even-aged stands led to stand stagnation in the 1980s, which acted as a predisposing factor for the climate change-induced forest decline symptoms reported since 1994–1995, associated with a series of very intense drought spells that acted as an inciting factor (Linares & Carreira, 2009). Finally, tree growth decline and loss of vigor led to the expansion of the root-rot fungus pathogen Heterobasidion abietinum (Linares et al., 2010), which acted as a contributing factor (Manion, 1981) causing widespread mortality and extensive formation of forest-gaps in the last two decades (>1/3 of the previous basal area lost). This multifactorial forest decline and dieback process increases the production of HR7 and R4 fuel models, as shown in Fig. 7. Under the prevalent “no-management” policy, these new open areas are, eventually, being invaded by dense shrubs, as supported by our LIDAR and ForeStereo data. This increases their importance in the fire behavior and promotes fuel models with high fire spread rate such as the UCO M9 fuel type. The fuel model classification revealed a remarkable contribution of M9 (Fig. 4), covering 30.5% of the study area. This suggests that shrub invasion is taking place and is already in an advanced phase. Also, the CBD values point to a high exposure to crown fires (Arellano Pérez et al., 2017) and could explain the forecasted high flame length in some areas.

As explained in the Introduction section, fire intensity in pinsapo forests is known to be low, but the above-mentioned current invasion into the mortality gaps by the surrounding dense shrubs could invert this tendency. However, it must be highlighted that the efficacy of employing fire simulations in risk management strongly depends on input data of high accuracy and precision, due to the complex heterogeneity of forest landscapes (Rodríguez y Silva & Molina-Martínez, 2012). Although we precisely determined shrub composition and structure in a set of training field plots, the low LIDAR point cloud density available hindered reliable mapping of understory vegetation, which thus may restrict the accuracy of the obtained fire risk simulations. The combined use of LIDAR, both terrestrial and airborne, could be the best option to map fuel models and canopy data such as Canopy Base Height and Canopy Bulk Density, for increased accuracy. Nevertheless, ForeStereo was shown to be a useful alternative to terrestrial LIDAR for calculating stand structure. Our study attempts to set a precedent as the first approach to fire risk analysis in Abies pinsapo forests using LIDAR. Also, it demonstrates the significant potential of this method for study of the ecological structure of populations of endangered fir species, and to broaden the understanding of their conservation status. Most of the current work with LIDAR data focuses on forests with commercial interest, and few studies have employed this technology to understand the structure of the populations of endangered species forests and their vulnerability to fire risk.

Conclusions

Our results show a high fire risk for the largest remaining continuous forest of the relict and endangered A. pinsapo tree species. Such risk, especially under east wind conditions (Levant), was found to be associated with a remarkable presence of shrub-dominated fuel models (M9). Using aerial LIDAR and ForeStereo data to assess stand structure spatial variability in the area, we found symptoms of stand stagnation and forest decline under the current no-management conservation policy. This process together with climate change trends triggers the formation of mortality-gaps that are eventually invaded by shrubs, increasing the production of the M9 fuel model, which in turn worsen fire risk. These findings stress the need for proactive adaptive management of A. pinsapo forests, including: (i) the creation of bare patches through shrub clearing, (ii) a reinforcement of the firewalls in the west part of the valley and (iii) promotion of grazing and trampling levels by wild ungulates (or domestic livestock if they were insufficient) to reduce shrub fuel load without compromising A. pinsapo. We also support the efficacy of thinning treatments for canopy structural diversity enhancement as an essential tool to avoid stand stagnation (Linares, Camarero & Carreira, 2009; Linares et al., 2009; Lechuga et al., 2017, 2019) and high CBD values, to reduce the probability of crown fires and thus increase resilience to wildfires (Koontz et al., 2020), as well as to reduce climate change-induced tree mortality (Linares, Camarero & Carreira, 2009).

The importance of the A.pinsapo populations in Sierra de las Nieves is one of the main reasons that inspired Spanish national policy to upgrade this protected area into a National Park. Although the models obtained have low accuracy due to technical limitations, our study provides a preliminary estimate, a first step to assess pinsapo forest risk factors using remote and proximal sensing as essential tools to support conservation management. These methods can also be extended to the monitoring of other endangered Western Mediterranean relict fir forests such as those of A. numidica and A. pinsapo marocana in North Africa and can be implemented in their conservation strategies.

Supplemental Information

Supplemental Information 1 Raw data.

Forestereo outputs, lidar outpus and additional shrubs data for fuel models.

Click here for additional data file.

We thank Víctor Lechuga and Antonio Román (University of Jaén), for assistance during fieldwork, Fernando Montes Pita, (INIA-CIFOR) for facilitating the ForeStereo device, which was essential for the fieldwork, and José Luis López Quintanilla (coordinator of the regional government plan for the restoration and conservation of Abies pinsapo habitats) for providing relevant information about the study area and species. Dr. Eric C. Henry kindly revised the English style and usage.

Abbreviations

CC Canopy Cover

CBD Canopy Bulk Density

CBH Canopy Base Height

G Basal area

Ho Stand height

KIA Kappa Index of Agreement

MMT Minimum Travel Time

RMSE Root-mean-square error

SD Standard Deviation

Additional Information and Declarations

Competing Interests

Author Contributions

Field Study Permissions

Patent Disclosures

Data Availability

Antonio Flores-Moya is a PeerJ Academic Editor.

Álvaro Cortés-Molino conceived and designed the experiments, performed the experiments, analyzed the data, prepared figures and/or tables, authored or reviewed drafts of the paper, and approved the final draft.

Isabel Aulló-Maestro conceived and designed the experiments, analyzed the data, authored or reviewed drafts of the paper, and approved the final draft.

Ismael Fernandez-Luque analyzed the data, prepared figures and/or tables, and approved the final draft.

Antonio Flores-Moya analyzed the data, authored or reviewed drafts of the paper, and approved the final draft.

José A. Carreira conceived and designed the experiments, performed the experiments, analyzed the data, authored or reviewed drafts of the paper, and approved the final draft.

A Enrique Salvo analyzed the data, authored or reviewed drafts of the paper, and approved the final draft.

The following information was supplied relating to field study approvals (i.e., approving body and any reference numbers):

Field experiments were approved by the “Consejería de Medio Ambiente y Ordenación del Territorio, Junta de Andalucía”. This is the Department of Environment and Land Management from the Andalusian Regional Government (Project number: PNSN/AU/10-2018).

The following patent dependencies were disclosed by the authors:

ForeStereo, based on MU2005-01738, patented by INIA-CIFOR in 2005

Patent disclosure: http://consultas2.oepm.es/InvenesWeb/detalle?p=1&referencia=U200501738.

The following information was supplied regarding data availability:

Raw data are available as a Supplemental File.

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
