# Peer review of "Using ForeStereo and LIDAR data to assess fire and canopy structure-related risks in relict Abies pinsapo Boiss. forests"

_PeerJ, doi:10.7717/peerj.10158_

## Round 0.1 · original submission · Major Revisions

I have received comments from two reviewers and both of them suggest "major revisions". Therefore, I invite you to submit a revised ms. Thank you!

·

Basic reporting

Here are my suggestions:
1. Many punctuation errors, phrase errors, and typos in the article need to be fixed. Free grammar and spelling check tools are available on the internet, they help identify wrong spellings and grammar issues, please go over your manuscript and check again.
2. Line 82. I suggest using a stronger mood than “might” to introduce RS application in your study.
3. Line 125 and 126 cite works of Linares, et al. in 2009, but these are two different articles 2009, please change them into 2009a and 2009b to tell them apart. Another works of Linares, et al. in 2011 also need to change.
4. Line132. 2.1 study place might be better if it changes into study area or study site.
5. Line 139. “…Paeonio broteroi-Abietetum pinsapo Asensi & Rivas-Martínez, 1976 composed…” should be “…Paeonio broteroi-Abietetum pinsapo (Asensi & Rivas-Martínez, 1976) composed…”.
6. Line 156. megapixels would be clearer than “Mp”.
7. Line 151 “1.2 Fieldwork: ForeStereo inventory” should be “2.2 Fieldwork…”
8. Line 177. The full name of the PONA project is needed.
9. Line 186. Citation of Scott & Burgan (2005) is listed in the reference.
10. Please change the symbology of the lower map in Fig 1 and leave the outline of the boundary, so the image can be seen clearly.
11. Table 1. The Italic-type description of fuel model categories needs to align altogether.
12. Table 2. The LFCC4, AR4, LFCCM, and LFCC0.25 are not explained in the article for its meaning and how to measure them.
13. Fig 3. It will nicer for readers to tell which plots are for training and validation, respectively, if different symbols or colors are applied for two groups of plots.
14. Fig 4. I see the contrast between east and west wind scenarios, but the difference would be more poping-out if both cases use the same levels from low to high. For example, when mapping the rate of spread, use a classification that encompasses slow speed on the left and fast speed on the right and apply colors from blue to red. The title of “Rate of Spread” on both sides are different. Please change them.
15. The second and third paragraph (line156-165) in 2.2 seems more adhere to the introduction of ForeStereo, so I suggest moving them to line 106 before the introduction of LiDAR.
16. The last paragraph of 2.1 Study place seems more adhere to the topic of fieldwork, I suggest moving it to the top of 2.2.
17. The error matrix of fuel model classification should be present in the article or supporting material or supplement.
18. Is there any measure besides cleaning shrubs that can be taken based on the simulation?

Experimental design

What were the metrics acquired by ForeStereo and how were they calibrated with field measurement by investigators? And What were the metrics from LiDAR used in regression? These metrics and procedures are not so clearly stated in the Material and Methods section that readers can’t figure out the analysis process until they read the results. These are especially important for future readers who do not know anything about the ForeStereo and the related procedures.

It is quite straightforward that the east wind causes worse scenarios because the topography seems to dominate in the fire environment. Will this depart from your result a lot when a different fuel moisture condition is applied? If that is the case, this simplicity (line 195-196) only demonstrates the very dry condition in this area, and if this condition is frequent in the study area, the result is convincing, otherwise, the rationale of this simplicity is needed.

Validity of the findings

You mentioned in the line 118-119 that the weak relationship between age and size. Is this because of declining growth? And your finding (in Fig 5) concert with their finding?

Reviewer 2 ·

Basic reporting

See Comments for the authors

Experimental design

See Comments for the authors

Validity of the findings

See Comments for the authors

Additional comments

GENERAL COMMENTS

The manuscript entitled “Using ForeStereo and LIDAR data to assess fire and canopy structure-related risks of Abies pinsapo Boiss forests” has the goal to combine fieldwork data with LIDAR information to characterize Abies pinsapo forests in a study area of southern Spain and to assess potential fire behavior and risk in the area.
The subject of the manuscript falls within the scope of the journal. Overall, the topic investigated is relevant and of great importance for Mediterranean fire-prone areas, particularly for biodiversity protection and forest management point of views.
The study cannot be considered original, as it combines previously developed methods already published in similar studies; nonetheless, it is an interesting work for the relevance of the Abies forests of the area.
I am not a big expert on ForeStereo and LIDAR data methodologies, so my comments will mostly focus on the fire risk part, in which several points need to be carefully revised or clarified.
The Introduction section should be enriched by adding some sentences and references to previous works that analyzed similar topics. In the present version, the fire risk part is absent from the Introduction. Furthermore, there is need to better focus the Introduction on the main goals of the work (“Using ForeStereo and LIDAR data to assess fire and canopy structure-related risks of Abies pinsapo Boiss forests”), that is some redundant parts or sentences could be omitted or shortened.
The Material and Methods section has some parts that should be better addressed and/or require additional explanations from the authors. Overall, the fire risk methods are not sufficiently complete to allow replication of the work, and the calibration of the FlamMap model is not evident. In other words, it seems that the authors used a model without any preliminary calibration and validation phase. The Results and Discussion sections need to be strengthened and revised, and should be improved taking into consideration the Specific Comments below.
The length of the text is not extended (less than 350 rows), and could be reinforced to clarify methods and results, as well as to better focus the Introduction section.
The use of English could be improved to increase the text fluency and to remove some minor grammatical wobbles.
I recommend a major revision of the manuscript before publication.



SPECIFIC COMMENTS

L35: Please consider using a more generic description of the methods instead of using “ForeStereo”. Alternatively, please describe in few words the mean characteristics/goals of “ForeStereo”

L62-135: The fire risk part is absent in the Introduction section, although the authors used a propagation model to characterize fire risk in the study area. There is need for providing at least the state of the art of the application of FlamMap or other models at the Spanish, European or Mediterranean level, as well as the potential limitations/strengths. In addition, the authors should justify the selection of FlamMap rather than other fire models (e.g.: FARSITE, Wildfire Analyst, etc.). At the EU scale, some suggested works are the following ones: Botequim et al. 2019 (https://www.publish.csiro.au/wf/WF19001), Alcasena et al. 2018 (https://www.sciencedirect.com/science/article/pii/S0301479718311551), Mallinis et al. 2016 (https://www.mdpi.com/1999-4907/7/2/46), Salis et al. 2016 (https://www.publish.csiro.au/wf/WF15081).
Likewise, please mention other previous works that (at least at the Spanish level) used remote sensing/LIDAR data to characterize surface and crown fuels and then inform fire spread/risk models.

L140: “The forest is placed in a steep valley of 252.59 ha”. I would suggest using “of about 250 ha”.

L137-157: The description of the “Study place” (or better “Study Area”?) section should be improved and modified.
The second paragraph (L149-157) should be merged with another section, because it contains methodological details rather than a description of the study area. The first paragraph (L137-148) could provide some other details, as for instance the main climate (e.g.: annual rain, winds, maximum temperatures in the summer season) and the fire regime of the study area.

L192-196: Please clarify in a sentence how the fuel models classification of Scott & Burgan was adapted to Southern Spain climate conditions

L202-204: Please provide the final resolution of the landscape file

L204-205: This is something that needs a clear explanation. As the authors know, fuel moisture files are mandatory files for FlamMap simulations, and so it is not possible to run FlamMap without these files. Please provide in a Table the moisture values associated with each fuel model for your simulations.

L205-207: Please specify if spot fires were simulated or not. If the answer is yes, please consider to provide the spot probability and spotting delay values used for FlamMap runs. Plus, please specify the simulation time (unlimited per each fire ignition, or how many hours of propagation?)

L226-228: The fire simulation results are not described. The reference to Figure 4 is not enough. The next lines (L244-255) are probably more appropriate in the Results part rather than in the Discussion.

L245-248: The authors reported that “Spread rate shows an average value of 54.85 m∙ min-1 in Levant and 22.85 m∙ min-1 in Ponent winds. More than 50% of the area is affected by a flame length above 30 m in Levant winds, whereas in Ponent winds this is only the 16.1% of the area”. The spread rate values are very high, particularly because wind speed was moderate (16 km h-1). For instance, 54.85 m min-1 corresponds to about 3.3 km h-1, that is a lot. How do the authors explain these values so high? Are these spread rates common in the surrounding mountain areas? The same consideration is valid for flame length, which is on average very high. Please provide your comments on this. Furthermore, please compare your fire behavior results with those reported in similar studies carried out in the Iberic Peninsula or in similar Mediterranean-climate areas, and improve the discussion of the study in the light of this comparison.

Figures 3 and 4. Please consider using easier classification values in the legend (e.g.: 20 rather than 21.4, 40 rather than 42.9, and so on). This could help readers. In Figure 4, please use the same scales when comparing BP, SR and FL values.

---

## Round 0.2 · Major Revisions

I received two sets of comments from the first round of review. Both of them were thorough, but comments from the anonymous one were more in-depth, and those from the other one were not too much. I returned your revision to both of the original reviewers but failed to obtain one that I was hoping to have. Therefore, I have to send your ms to a third reviewer before making my final decision. The third reviewer raises some critical issues, which I concur with them. On top of those, I feel that the Introduction section is way too lengthy (starting from L108) and should be downsized substantially. Therefore, a “major revisions” tag is given for this revision.

·

Basic reporting

No comment.

Experimental design

No comment.

Validity of the findings

No comment.

Additional comments

For a better manuscript, please check again the punctuation, typo, and redundant words with Grammarly before publishing. Some instances are listed, including but not limited to these.
1. Line 42. "With this information',' we developed regression models..."
2. Line 49. "...with 'the' potential for high fire spread rate fire and burn probability."
3. Line 52. circunmediterranean or circum-Mediterranean?
4. Line 77 ."...experienced an outstanding speciation...". Please delete 'an'.
5. Line 79. "Past climate changes have led to population migrations, and to shrinkage and fragmentation..."
6. Line 93. "...detection of forest infestations (Immitzer & Atzberger, 2014) to estimate evapotranspiration..."
7.Line 97. "... 'the' point cloud..."

Reviewer 3 ·

Basic reporting

The article must be use clear, unambiguous, technically correct text.

Experimental design

Methods should be described with sufficient information to be reproducible by another investigator.

Validity of the findings

Speculation is welcome, but should be identified by proving.

Additional comments

General Comments
This study may contribute to existing literature on “combine ForeStereo and LIDAR data to assess fire in endangered fir forests”. Research objectives are provided a new procedure to map fire risk and canopy structure spatial variability. The overall level of the paper is good, but there have some questions need to be clearly demonstrated the methods and purpose of the research. The study used ForeStereo and LiDAR data for characterizing the forest structure. The author should be noted that two methods might have measurement error, due to they focused different scale and had unequal variance. There is lack of the results to prove the consistency of two methods in the manuscript. I believe that the MS needs some clarifications and improvements before being considered for publication. A few additional comments are given below:

1. The author did a lot of analysis works and output results, but did not discussed tables and figures completely, e.g. Fig 3, Fig 4, Fig 5, table 2 etc. I suggest that author don’t just report data. The author should be good to have more discussion about the generalization of the research findings.

2.The figure captions is too simple. e.g. Fig 5, What is A and B?? I will recommend the author should be described figure captions more details and more information.

3.L206-209. The author has sampled 49 plots for obtaining tree metrics and for classifying the fuel models. This number of plots is enough for statistics analysis. The author didn’t mention why fixed the train model plots (yellow points) and validation plots (red ones). I will recommend splitting a dataset randomly into training and test datasets divided it into smaller sets for building up and validating a model.

4. L277-281. The author used Scott & Burgan (2005) suggestions to distinguish initial fuel moisture. The author didn’t mention what meaning of shaded spots (M8, HR7, and R4) , very low content (fully cured) and sunny spots (M3, M9 and HPM4).

5.L244, The author says the LiDAR pixel size is 0.25 m and the point cloud density is 0.5 points∙m-2 in ms. It means that 1 m2 could not rasterize to 1 pixel. Please check it.

6. L309-L315, The author showed that results of LIDAR metrics and ForeStereo data. The model fitting is not so good. The Canopy Cover (CC) had the best fitting model
with an RMSE less than 20%, but others showed great variance. The results might lead to failure analysis and inference. The limitations and the applicability of this work to other regions should also be discussed.

7. The ms is not easy to read. I will recommend revising the ms structure.

Additionally, I’ve found some minor concerns:
1.Table 1 , Table 1 and Table 3. please unify the digit after the decimal point in the manuscript.
2. Table 2, Please correct “tn” to “ton”
3. Please unify Kappa Coefficient (L46) and KIA
4. The predictive model formula in Table 4 should be use Mathematical equation format.

---

## Round 0.3 · Minor Revisions

Please see the comments. You should comply with them and make the necessary changes. I will make a final decision in the next round.

Reviewer 3 ·

Basic reporting

The article must be written clearly in method section.

Experimental design

Research question well defined, relevant & meaningful.

Validity of the findings

All underlying data have been provided.

Additional comments

General Comments
This is a potentially very interesting article. The treated topic is actual and very important for assessing fire and canopy structure-related risks. It discussed " a forest inventory device, to assess fire risk and canopy structure spatial variability by using aerial LIDAR and hemispherical images". The Response letter line numbers are didn’t matching with revision MS. The author should notice next time. But, the revision appropriately addressed many of the comments raised by the reviewers. However, I believe that the MS needs some clarifications and improvements before being considered for publication. I haven't big issue to raise, only few moderate-major points. A few additional comments are given below:

1. L204-205. The author had splitted randomly into train and validation, but Figure 1 still shown Yellow points are fitting plots, and red ones are validation plots. Please check it.

2.L184, The 49 plots are used 8 m radius, and author says the LiDAR point cloud density is 0.5 points∙m-2 in ms. Please specify accuracy of GPS device. The author mention that some high difference could be due to low point cloud density (minimum 0.5 points/m2 guaranteed) as well as the fact that airborne LIDAR produces better accuracy for variables related to the top of the canopy (Hilker, et al., 2012). Why canopy cover has high R2. Please specify it.

3. My second suggestion is about the regression model (Table 3). There should be mention that method of model fit and scatterplots to show relationship derived predictive model on y-axis and field collected data on the X-axis. The model method is incomplete, insufficient and incorrect. A complete description should provide how to model fitted. I would like to know how LiDAR attributes are associated with field drive observations. Please add the statistical analysis.

Additionally, I’ve found some minor concerns:
1. The aspect 0° and 360° should be the same. Please check it.
2. Table 2, the PP(mm) and cloud(%) are all zero, please addressing in 2.2 Fieldwork subsection.

---

## Round 0.4 · accepted · Accept

I am pleased to inform you that the quality of the revision is satisfactory. I have recommended the acceptance of this paper. Good job!